# Determination of Mercury in Fish Sauces by Thermal Decomposition Gold Amalgamation Atomic Absorption Spectroscopy after Preconcentration by Diffusive Gradients in Thin Films Technique

**DOI:** 10.3390/foods9121858

**Published:** 2020-12-12

**Authors:** Pavel Diviš, Marek Reichstädter, Yue Gao, Martine Leermakers, Jakub Křikala

**Affiliations:** 1Faculty of Chemistry, Brno University of Technology, Purkyňova 118, 612 00 Brno, Czech Republic; xcreichstadter@fch.vut.cz (M.R.); xckrikala@fch.vut.cz (J.K.); 2Analytical, Environmental and Geo-Chemistry, Vrije Universiteit Brussel, Pleinlaan 2, 1050 Brussels, Belgium; Yue.Gao@vub.be (Y.G.); mleermak@vub.be (M.L.)

**Keywords:** mercury, extraction, preconcentration, fish sauce, diffusive gradients in thin films, atomic absorption spectroscopy

## Abstract

The analysis of mercury in food presents a challenge for analytical chemists. Sample pre-treatment and the preconcentration of mercury prior to measurement are required, even when highly sensitive analytical methods are used. In this work, the Diffusive Gradients in Thin Films technique (DGT), combined with thermal decomposition gold amalgamation atomic absorption spectrometry (TDA-AAS), was investigated for the determination of the total dissolved mercury in fish sauces. Moreover, a new type of binding gel with Purolite S924 resin was used in DGT. Linearity assays for DGT provided determination coefficients around 0.995. Repeatability tests showed a relative standard deviation of less than 10%. pH values in the range of 3–6, as well as NaCl concentrations up to 50 g·L^−1^, did not affect the performance of DGT. The effective diffusion coefficient of mercury in five-fold diluted fish sauce was determined to be (3.42 ± 0.23)·10^−6^ cm^2^·s^−1^. Based on the 24 h deployment time of DGT, the limit of detection (LOD) for the investigated method was 0.071 µg·L^−1^. The proposed method, which combines DGT and TDA-AAS, allows for the analysis of fish sauces with mercury concentrations below the LOD of TDA-AAS, and significantly reduces the wear and corrosion of the TDA-AAS components.

## 1. Introduction

Fish sauce is an amber-coloured liquid that is produced by fermenting fish with sea salt and is used as a flavouring, mainly in South East Asian cuisine [1]. Fish sauce is a food with a complex chemical composition, mainly influenced by the type of fish used for production and by the fermentation conditions. The main components of fish sauce are amino acids, organic acids, major elements, and water-soluble vitamins [2]. The typical characteristics of fish sauces are high salt content, ranging up to 300 g·L^−1^ [3].

Aquatic systems are subjected to pollution pressures associated with population growth, urbanisation and industrialisation. Pollutants in aquatic systems can be eithter organic or inorganic, including hazardous metals. Fish can accumulate those hazardous metals through gill, surface skin and oral ingestion of food and particulate material suspended in water. The bioaccumulation of hazardous metals in fish can lead to potential health risks to humans through fish product consumption. One of the most monitored hazardous metals in fish products is mercury (Hg). Chronic Hg poisoning causes hair loss, digestive disorders, neurological and psychological problems, and other serious health problems, such as anaemia, rheumatic diseases, and kidney damage. Chronic exposure to Hg can also cause tooth decay, rashes, muscle tremors, memory loss, behavioural changes, and central nervous system damage [4]. Therefore, to protect consumer health, it is important to control the amounts of Hg in foods, a task which requires sensitive methods for analysing Hg levels in food products. The maximum permitted Hg content in fish products is determined by Commission Regulation (EC) No. 1881/2006 [5].

The most common methods used to determine total Hg in various types of samples are spectroscopic methods, including cold vapour atomic absorption spectrometry, cold vapour atomic fluorescence spectroscopy, and inductively coupled plasma mass spectrometry [6]. A cost-effective alternative to these conventional techniques is thermal decomposition gold amalgamation atomic absorption spectrometry, which additionally allows for the direct determination of Hg in solid matrices [7]. Due to the low concentration of Hg in many types of samples and complicated matrices, causing suppression or the enhancement of the analyte signal, sample pre-treatment [6], preconcentration, and separation of Hg prior to measurement are required, even when highly sensitive analytical methods are used [8]. The most widely used techniques for the separation and preconcentration of Hg at low concentrations are classic extraction techniques [9].

In 1994, the Diffusive Gradients in Thin Films technique (DGT) was introduced by Davison and Zhang. This technique has greatly simplified the determination of zinc in seawater, as it effectively separates the analyte from the matrix, which otherwise causes specific problems during analysis [10]. The principle of DGT has been described extensively in the literature [11,12]. Although DGT was originally designed for environmental analysis, several other applications of DGT are currently being developed [13,14]. At present, in combination with instrumental methods, more than 50 elements can be determined by DGT [15]; studies have also been performed, exploring its use in Hg determination [16,17,18,19,20,21,22]. The major problem encountered when using DGT to determine Hg levels is that most of the previously tested Hg-selective sorbents are no longer commercially available; only 3-mercaptopropyl silica gel particles are currently available to assess Hg by DGT.

This work focuses on the determination of Hg in food seasonings using the DGT with Purolite S924 binding gels. Purolite S924 is a polystyrene-based resin with thiol functional groups, which could be used as a possible replacement for currently used binding gel containing 3-mercaptopropyl silica gel particles. The need to develop DGT application capabilities for analysis of high salinity condiments was mentioned in a recent study of Chen et al. [14], investigating the application of DGT for the determination of lead in soybean sauce. At present, however, the application of the DGT in food analysis has not been explored, although it offers exciting options for separating the analyte from a complicated matrix of food samples and provides several other benefits in the analytical process. The aim of this work was the validation of DGT with Purolite S924 binding gel and verification of the possibility of using this technique in combination with thermal decomposition gold amalgamation atomic absorption spectrometry for the determination of total dissolved Hg in fish sauce samples.

## 2. Materials and Methods

### 2.1. Chemicals, Solutions, and other Material

Mercury standard solution (Astasol, 1000 mg·L^−1^, Analytika, Prague, Czech Republic) was used to prepare all model solutions containing different concentrations of Hg. Nitric acid (Analpure, Analytika, Prague, Czech Republic) and sodium hydroxide (Penta, Prague, Czech Republic) were used to adjust the final pH of the model solutions. Sodium chloride (Penta, Prague, Czech Republic) was used to prepare solutions with different salt content. Purolite S924 chelating resin with thiol functional groups (Purolite, Paris, France) was used to prepare the binding gel. Since the mean size of these resin particles was 0.5–0.8 mm, an unsatisfactory size for preparing thin, compact binding gels, the resin was milled to an average particle size of 0.1 mm and sieved through a 0.1 mm sieve prior to use. The binding gel was then prepared by mixing 0.35 g of Purolite S924 resin with 4 mL of hot 1.5% agarose (Merck, Darmstadt, Germany) solution and pipetting the mixture between two glass plates that were separated from each other using 0.05 cm thick Teflon foil. The diffusive gel was prepared in the same way by only using 1.5% hot agarose solution. To prepare diffusive gels with thicknesses other than 0.05 cm, Teflon foils with thicknesses of 0.025, 0.075 and 0.1 cm were used. The empty DGT pistons were purchased from DGT Research Ltd. (DGT Research, Lancaster, UK) and assembled as described by Docekalova and Divis [16]. The layer of binding gel and diffusive gel was covered with a membrane filter (Pall, New York, NY, USA, diameter 2.5 cm, thickness 0.014 cm, pores 0.45 µm). The basic model solutions and spiked 5-fold diluted fish sauce solutions were all prepared as 4 L (basic model solution) or 2 L (5-fold diluted fish sauce) solutions containing 25 µg·L^−1^ Hg^2+^. In addition, the basic model solutions contained 0.01 mol·L^−1^ NaCl (Penta, Prague, Czech Republic). In order to test the influence of different pH and salt content, these properties of the basic solution were modified. Final DGT performance testing was conducted on a 5-fold diluted fish sauce spiked with 5 µg·L^−1^ Hg^2+^. All water used in this study was ultrapure water (Elga Purelab, High Wycombe, UK, resistivity 18 MΩ·cm). All solutions were stirred at 400 rpm using a magnetic stirrer (IKA, Staufen, Germany). Temperature and pH measurements during the experiments were performed using a pH probe (WTW GmbH, Weilheim, Germany). Fish sauce is a sample with a complex matrix which can be classified as either a solution or a suspension, and there is no certified material on the market that mimics the composition of fish sauces. Therefore, several different samples were used in the validation of thermal decomposition gold amalgamation atomic absorption spectrometry, despite the fact that this instrument provides results which are almost matrix independent. As a liquid sample, a wastewater sample (PT/CHA/2/2018, CSlab, Prague, Czech Republic) was analysed within an external quality assessment. Fish muscle meat sample ERM-BB422 (Sigma-Aldrich, Darmstadt, Germany) with certified mercury content 0.601 ± 0.030 mg·kg^−1^ was chosen as the primary sample for the validation study and a sediment sample ERM-CC580 (Sigma-Aldrich, Darmstadt, Germany) with a certified mercury content of 132 ± 3 mg·kg^−1^ was used to verify the correct operation of the instrument on a daily basis.

### 2.2. Instrumentation

An Advanced Mercury Analyser AMA 254 (Altec, Prague, Czech Republic) was used for Hg analysis in the model solutions, binding gels, and commercially available fish sauce samples. This instrument is principally based on combustion of the sample in a combustion chamber flooded with oxygen. The released gases were removed from the combustion chamber and passed over a gold amalgamation trap to separate the atomic Hg. The amalgamation cell was heated to release the Hg, which was finally measured by atomic absorption spectroscopy [23]. The working program of the Advanced Mercury Analyser consisted of 60 s of drying, 150 s of decomposition and 45 s of waiting. The Advanced Mercury Analyser is further referred hereto as thermal decomposition gold amalgamation atomic absorption spectrometry, abbreviated as TDA-AAS.

### 2.3. Method Validation

The validation of TDA-AAS was performed via evaluations of the limit of detection (LOD), the limit of quantification (LOQ), trueness, and precision. The LOD of TDA-AAS was calculated from the average background signal obtained from 10 parallel analyses of 100 µL ultrapure water and raised by three times the standard deviation (SD). The LOQ of TDA-AAS was calculated from the average background signal obtained from 10 parallel analyses of 100 µL ultrapure water and raised by ten times the SD. Trueness and precision were evaluated by analyses of ERM-BB422, a certified reference material. The long-term stability of the TDA-AAS performance was evaluated through analysing the certified reference material ERM-CC580 every day for the duration of the experimental period leading up to sample analysis and a control chart was generated from these measurements. Additionally, TDA-AAS performance was further validated through participation in an interlaboratory trial (PT/CHA/2/2018), in which samples of wastewater were analysed.

The validation of DGT was performed both using a model solution and a diluted fish sauce solution. The linearity of Hg accumulation in the binding gel over time, the value of the effective diffusive coefficient (*D_e_*), the thickness of the diffusive boundary layer (DBL, *δ*), and the performance of DGT in solutions with different pH and NaCl concentrations were all evaluated, along with the LOD, LOQ, trueness, and precision. LOD was calculated from the average mass of Hg in binding gel blanks raised by three times the SD and by taking into account the determined effective diffusion coefficient, the standard parameters of the DGT unit, and the exposure time of 24 h, according to Equation (1), where *M* = mass of analyte; Δ*g* = diffusive layer thickness; *D_e_* = effective diffusion coefficient of analyte in the diffusive gel; *t* = deployment time; *A* = exposure area.
*c*DGT = (*M*·Δ*g*)/(*D_e_*·*t*·*A*)(1)

The LOQ was calculated in a similar way using the average mass of Hg in the binding gel blanks raised by ten times the SD. The trueness of DGT was evaluated using the recovery for a 5-fold diluted fish sauce sample spiked with a known amount of Hg. The precision of DGT was assessed by calculating the relative standard deviation (RSD) of DGT concentrations calculated from Equation (1) in the analysis of 6 parallel samples of binding gel peeled off from DGT pistons at the end of the deployment time in 5-fold diluted fish sauce.

The effective diffusion coefficient, *D_e_*, was calculated after performing tests on the linear accumulation of Hg during the exposure time. Ten DGT pistons were immersed in the model solution and two units were removed from the solution after 2, 4, 6, 8, and 24 h. The DGT pistons were then dismantled and the binding gel was analysed using TDA-AAS. By plotting the dependence of *M*.*c*^−1^ on time, the slope *k* of linear regression was obtained and *D_e_* was calculated using Equation (2) [16], where *t_s_* is the conversion factor of hours to seconds.
*D_e_* = *k*·Δ*g*·*A^−^*^1^·*t_s_*^−1^(2)

This procedure was also used to calculate the *D_e_* of Hg in diluted fish sauce. The *D_e_* can be affected by the uptake efficiency of the analyte on the binding gel; therefore, it is essential to verify uptake efficiency, especially in complex matrices. In this work, we adopted our procedure for determining the uptake efficiency of Hg on the binding gel from the work of Abdulbur-Alfakhoury et al. [24].

The thickness of the DBL was calculated from the simultaneous deployment of eight DGT pistons with different thicknesses of diffusive layer (diffusive gel and membrane filters 0.039, 0.064, 0.089, and 0.114 cm, tested in duplicate) in the model solution. After plotting *M*^−1^ against Δ*g*, the DBL was calculated from the slope k and intercept *q* obtained from the regression equation of dependence of *M*^−1^ to Δ*g* [25]. The diffusion coefficient of Hg in water, *D_w_*, was taken from the work of Docekalova and Divis [16].
*δ* = *q*·*k*^−1^·(*D_w_/D_e_*)(3)

All experiments were performed at the laboratory temperature of 20 °C and were repeated three times.

The effect of pH of the model solution of DGT performance was evaluated by immersing four pistons into the model solution with a pH of 3 for 4 h. This test was then repeated in model solutions adjusted to pH 4, 5, and 6. The concentration of Hg in the model solution was monitored for the duration of the test by analysis of TDA-AAS and recorded as *c*SOL. This concentration was then compared with the concentration determined by DGT (*c*DGT) and recovery *R* was calculated as *R* = (*c*DGT/*c*SOL) × 100. To evaluate the effects of NaCl concentration of the solution on DGT performance, DGT was used on model solutions with NaCl concentration, ranging from 2–50 g·L^−1^, corresponding to the salt concentration in five-fold diluted fish sauce. The test arrangement was similar to that used in the testing of pH influence on DGT performance, and the *R* value was calculated again.

### 2.4. Analysis of Fish Sauce Samples

Samples of fish sauce were purchased at Asian markets in Brno, Czech Republic, and in Brussels, Belgium. Fish sauces were diluted with ultrapure water to a volume ratio of 1:4 and stirred for a total of 1 h in a 2 L glass beaker. Subsequently, four DGT pistons were inserted into the solution for a total time of 24 h. After the exposition, DGT units were dismantled, and the binding gels were directly analysed using TDA-AAS. The DGT concentration was calculated using Equation (1). The concentration of Hg in all undiluted samples was measured directly at the same time in triplicate by TDA-AAS. Concentrations were obtained in µg·L^−1^ (from both methods) and recalculated to the average concentrations (µg·L^−1^) and then recalculated to mass concentrations in mg·kg^−1^ using the average fish sauce density of 1.2 g·mL^−1^ [1]. All experimental data were statistically processed using XLstat software (Addinsoft, New York, NY, USA).

## 3. Results and Discussion

### 3.1. Validation of TDA-AAS

The values for the LOD and LOQ of TDA-AAS were determined to be 0.068 ng Hg (0.68 μg·L^−1^) and 0.13 ng (1.3 μg·L^−1^), respectively. The LOD of new TDA-AAS (AMA 254), as stated by the manufacturer, is 0.01 ng Hg. Most studies using this type of instrument for determination of Hg report this LOD (0.01 ng) without further verification [26,27,28]. Gao et al. [29] reported an LOD of 0.03 ng Hg for TDA-AAS; however, in our study, such an LOD could not be achieved through repeating the instrument’s cleaning program or replacing all of the consumables (seals, boat, boat holders). Thus, the LOD determined in this study is more realistic for a normally loaded device. The LOD of TDA-AAS should be sufficient for the analysis of Hg in most fish sauces; however, samples of fish sauces containing Hg concentrations below the LOD of TDA-AAS were also analysed in this study. Thanks to the possibility of direct analysis of fish sauce samples, the TDA-AAS technique used in this study provides a better LOD than inductively coupled plasma mass spectrometry, for which the decomposition of a complex matrix of fish sauce is required [30].

Trueness and precision of TDA-AAS were verified through analyses of the ERM-BB422 reference material. The difference between the determined and declared concentrations in the certified reference material was not significant (determined Hg concentration = 0.589 ± 0.017 mg·kg^−1^, n = 6, *p* = 0.3808, α = 0.05). The coefficient of variation calculated from the obtained results was 2.9%. Based on the measured results, it can be concluded that the TDA-AAS used in this study meets the performance requirements for analytical methods set by the Commission Decision 2002/657/EC [31].

The capability of TDA-AAS to provide accurate and reliable results within permissible levels of uncertainty was also tested by participating in an interlaboratory comparison (during which a sample of wastewater with a reference value of 5 µg·L^−1^ was analysed) and by the evaluation of the long-term behaviour of TDA-AAS. The achieved Z-score in an interlaboratory comparison, Z = 0.15, did not exceed the limit for satisfactory results (|Z| ≤ 2). To evaluate the long-term behaviour of TDA-AAS for Hg determination, control charts of Hg contents were obtained through the analysis of ERM-CC580 certified reference material. During all the analyses performed on the ERM-CC580 certified reference material, no out-of-control signal was measured proving the good stability of the analytical system. The difference between the determined average concentration (130.8 ± 2.6 mg·kg^−1^) and the declared concentration (132 ± 3 mg·kg^−1^) of Hg in the certified reference material was not significant (*n* = 53, *p* = 0.0880, *α* = 0.05). The summary results of the TDA-AAS validation are given in Table 1.

### 3.2. Validation of DGT

Standard DGT testing procedures include time series accumulation experiments, during which the linear accumulation of the analyte on the DGT binding gel is monitored as a function of time [11]. In the model solution, the mass of accumulated Hg in the binding gel increased linearly with increasing exposure time (*r*^2^ = 0.9991, Figure 1), while the RSD for DGT deployment was <10%, confirming proper DGT function. From the time series experiment, the effective Hg diffusion coefficient *D_e_* was calculated using Equation (2). The calculated *D_e_* of Hg in the model solution was (6.23 ± 0.19)·10^−6^ cm^2^·s^−1^. This value is lower than the diffusion coefficient of Hg in water [16]; however, values of *D_e_* lower than *D_w_* have already been described in NaCl solutions, owing to the formation of [Hg(Cl)_x_]^2−x^ complexes [18,20].

When DGT is immersed in a solution, a region is created close to the exposition window of the DGT piston where the transport of metal ions and complexes undergo a rapid transition from advective to diffusive control. This region is known as the diffusive boundary layer (DBL). The thickness of the DBL may bias the determined concentrations if it is not considered in the calculations and, therefore, should be verified with each application of DGT [25]. The higher the DBL in the solution due to insufficient mixing of the solution, the less analyte that will accumulate in the binding gel in the same period. To estimate DBL thickness in DGT, several different procedures can be used [32]. In this study, DBL was estimated using the inverse plot method. The inverse of the Hg mass was linearly correlated with Δ*g* (*r*^2^ = 0.9956, Appendix A), which is consistent with the theory of DGT. Under the experimental conditions in this study, the DBL thickness was calculated to be 50 µm. In other studies, DBL measurements in experiments ranged from 30 to 200 µm [11,25,30,32]. In this study, the pistons were mounted on a nylon cord (no plastic holder was used) and deployed in a larger volume of solution, which, together with intensive mixing, seems to have led to a lower DBL thickness than those published by other authors. Although the determined DBL thickness is small compared to the standard diffusion gel thickness (0.05 cm), this thickness was included in the DGT calculations in the form of extended Δ*g*.

The parameter *c*DGT can be influenced by changes in the pH of the solution. Changes in pH can affect the speciation of the analyte in solution and, thus, can affect the ability of the analyte to bind to the binding gel [33,34]. The pH of the solution may also change the structure of the functional groups of the resin used in the binding gel [35], thus influencing the performance of DGT [11,36]. The speciation of the analyte in the solution is also affected by the presence of inorganic or organic ligands in the solution. In the case of Hg, the effect of humic substances and chlorides on the performance of DGT is frequently monitored. Low DGT performance using a binding gel containing Chelex 100 resin to determine Hg content in the presence of chlorides or humic substances has already been described in the literature [17,21]. The performance of the DGT in solutions of different compositions is commonly performed by the investigation of R value. In this study, good agreement between *c*DGT and *c*SOL was obtained over the pH range investigated, resulting in R values between 93% and 98% (Figure 2). Because the typical pH value of fish sauce is approximately 5 [3], it can be concluded that DGT is suitable for application in this matrix. Due to the fact that fish sauces contain high levels of salt, DGT performance in solutions containing NaCl concentrations up to 50 g·L^−1^ was also investigated. As seen in Figure 2, high salt content solutions (at levels corresponding 5-fold diluted fish sauce) did not have a major impact on the performance of DGT.

Tests in the model solution were performed in order to compare DGT performance with other studies. However, the tests had to be repeated in a diluted fish sauce solution as this solution has a completely different composition. Plots of the inverse Hg mass versus time gave good correlation coefficients in the diluted fish sauce solution (*r^2^* = 0.9992, Figure 1). From the measured slope of the linear regression line, the *D_e_* of Hg in diluted fish sauce was calculated to be (3.42 ± 0.23)·10^−6^ cm^2^·s^−1^ (Equation (2)). This calculated *D_e_* is lower than *D_e_* values determined during the tests on model solutions (6.23·10^−6^ cm^2^·s^−1^), a finding which can be explained by the presence of substances other than NaCl diluted in the fish sauce solution, especially proteins and amino acids [2] that can bind Hg to their structure. To ensure that the *D_e_* of Hg is affected only by a change in Hg species in diluted fish sauce and not by decreases in adsorption efficiency, the uptake efficiency was verified in the model solution. As shown in Figure 3, the uptake of Hg from diluted fish sauce by the binding gel is a continuous and fast process, such that the influence of *D_e_* by adsorption processes can be avoided.

The LOD and LOQ of DGT were 0.071 µg·L^−1^ and 0.116 µg·L^−1^, respectively. These values are comparable to the LODs and LOQs achieved using DGT with other binding gels containing Hg-selective resins and TDA-AAS as an instrumental method [17], as well as to methods requiring the elution of Hg from the binding gel, such as inductively coupled plasma mass spectrometry and liquid chromatography coupled to cold vapour atomic fluorescence spectrometry methods [37,38]. Compared to direct analysis of fish sauce on TDA-AAS, LOD ten-times lower were achieved when preconcentrating the Hg by DGT.

Since no certified reference material corresponding to the composition of dilute fish sauce was available at the time of this study, trueness was evaluated using a recovery test. The background concentration of Hg in the fish sauce used for the test was 2.5 ± 0.3 µg·L^−1^. After dilution and spiking, the final concentration of Hg in the fish sauce was 5.6 ± 0.3 µg·L^−1^ (measured directly by TDA-AAS). The determined DGT concentration of Hg in the spiked 5-fold diluted fish sauce was 5.3 ± 0.4 µg·L^−1^ (*n* = 6), which corresponds to 95% recovery. The precision of DGT expressed as RSD was 5.6%. These results pass the standards recommended by DGT Research Ltd., indicating good accuracy by DGT utilising binding gel with Purolite S924 resin. The summary results of the DGT validation are given in Table 1.

### 3.3. Analysis of Fish Sauce Samples

After all validation testing, DGT was used to extract Hg from 10 different fish sauces (originating in Vietnam and Thailand) and, through TDA-AAS analysis of binding gels from DGT, the Hg concentrations of the fish sauce samples were calculated. Mercury concentration was also determined in parallel by the direct analysis of fish sauces by TDA-AAS. The analytical results are summarised in Table 2. Through the direct analysis of fish sauce samples using TDA-AAS, it was possible to determine the concentration of Hg in eight of the 10 samples of fish sauce. The concentration of Hg in two of the samples was below the LOD of TDA-AAS. The content of Hg in the fish sauces ranged from 0.8 to 42.8 µg·kg^−1^ and none of the samples examined exceeded the maximum authorized concentration of 500 µg·kg^−1^ that is set for fish products [10]. Mercury content in fish sauces is difficult to find in the literature. Mabesa et al. [39] and Funatsu et al. [40] reported concentrations of Hg in different types of fish sauces ranging in 1–90 µg·kg^−1^, which is consistent with the concentrations found in this work. The differences between Hg concentrations determined by direct analysis of the sample using TDA-AAS and by analysis of the sample using TDA-AAS after a DGT extraction step were not significant for all samples (*p* > 0.05).

Although it is possible to use TDA-AAS for direct analysis of fish sauce samples, the complicated salt matrix of the analysed samples produces a heavy burden on the components of the TDA-AAS and may eventually damage the catalyst of the instrument. Corrosion of the metal parts of the device is observable after only a few samples have been analysed. Some of the biggest advantages of the validated DGT are the preconcentration of Hg from a sample of fish sauce, which allows concentrations under the LOD of TDA-AAS to be determined, and the ability to separate Hg from the complex matrix of fish sauce, which results in longer service intervals of TDA-AAS. Thanks to the preconcentration ability of DGT, the Hg concentration in all 10 fish sauce samples could be determined.

### 3.4. Concern for Public Health

To evaluate the potential hazards resulting from the consumption of food containing Hg, the concept of an estimated daily intake (EDI) and estimated weekly intake (EWI) is commonly used [41,42]. Information on the amount of fish sauce consumed in the world is difficult to find; however, for the Asia–Pacific region, fish sauce consumption is reported as the highest, with a consumption of 4 kg per year per capita reported. If the average fish sauce consumption at a rate of 4 kg·year^−1^, average concentration of Hg in fish sauce found in this study 11.4 µg.kg^−1^ and body weight (b.w.) of 60 kg are taken into account, the calculated EDI is 0.0021 µg.kg^−1^ b.w. By multiplying EDI by a factor of seven, corresponding to 7 days, the EWI value is 0.014 µg.kg^−1^ b.w. Calculated EWI can be compared with the provisional tolerable weekly intake (PTWI) of 4 µg Hg.kg^−1^ b.w. set by the Joint FAO/WHO Expert Committee on Food Additives. Based on a comparison of these values it can be concluded that the consumption of fish sauce does not pose a significant health risk to humans. Even if we take into account the highest measured concentration of Hg in fish sauce in this study (42.8 µg.kg^−1^), it can still be stated that the exposure of Hg by normal ingestion of the fish sauce is negligible (EWI = 0.055 µg.kg^−1^ b.w); however, a wider range of samples need to be analysed to provide a more accurate estimate of the health risks associated with consumption of fish sauce.

## 4. Conclusions

In this work, DGT with Purolite S924 binding gel and agarose diffusive gel were successfully used in combination with TDA-AAS in the determination of total dissolved Hg in fish sauce. All validation parameters of the investigated method were within acceptable limits, indicating that the method has good accuracy and sensitivity. The results of this study provide a gentle and sensitive method for the determination of total dissolved Hg in complex high salt matrices, such as fish sauce.

## Figures and Tables

**Figure 1 foods-09-01858-f001:**
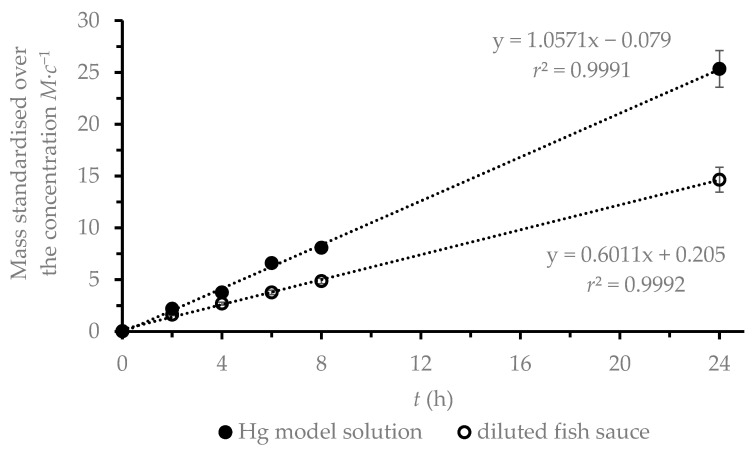
Measured mass of mercury in the binding gels immersed in Hg model solution and in diluted fish sauce over time.

**Figure 2 foods-09-01858-f002:**
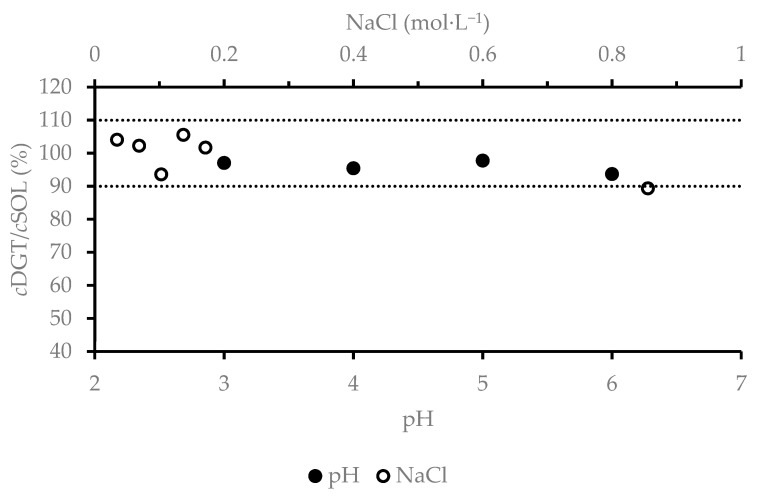
Influence of pH and NaCl concentration on recovery. RSD < 10 %.

**Figure 3 foods-09-01858-f003:**
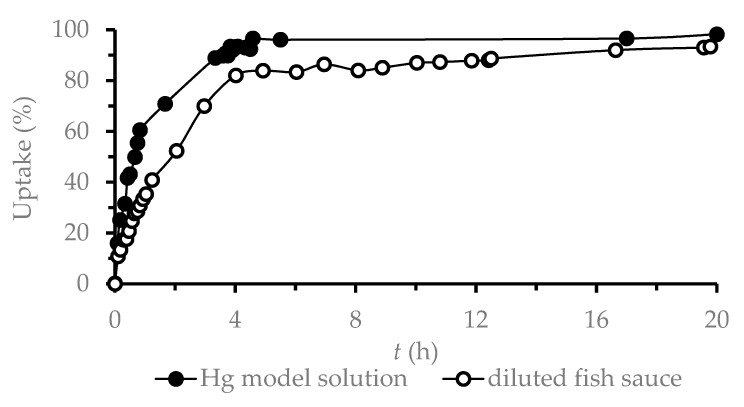
Uptake efficiency of Hg (1 mg·L^−1^) on the binding gel.

**Table 1 foods-09-01858-t001:** Results from method validation.

Parameter	TDA-AAS	DGT
LOD	0.68 µg·L^−1^	0.071 µg·L^−1^
LOQ	1.3 µg·L^−1^	0.12 µg·L^−1^
Relative bias (RE)	2.0%	5.4%
Repeatability (RSD_r_)	2.9%	5.6%
Reproducibility (RSD_R_)	4.3%	8.2%
Uncertainty (U)	8.6%	16%

U = *k*·RSD_R_, *k* = 2 at the 95% confidence level. TDA-AAS: thermal decomposition gold amalgamation atomic absorption spectrometry; DGT: Diffusive Gradients in Thin Films technique; LOD: limit of detection; LOQ: limit of quantification.

**Table 2 foods-09-01858-t002:** Concentration of Hg in fish sauce samples measured by direct analysis (*c*TDA-AAS) and by application of DGT (*c*DGT).

Sample No.	Sample Information	*c*DGT µg·L^−1^	*c*TDA-AAS µg·L^−1^	*c* µg·kg^−1^
FS1	Vietnam, anchovy content 95%	5.2 ± 0.4	4.5 ± 0.3	4.1
FS2	Vietnam, anchovy content 97%	3.0 ± 0.3	2.5 ± 0.3	1.3
FS3	Vietnam, anchovy content 70%	2.3 ± 0.2	1.9 ± 0.3	1.8
FS4	Thailand, anchovy content 70%	1.3 ± 0.1	<LOD	1.1
FS5	Thailand, anchovy content 77%	2.0 ± 0.2	1.6 ± 0.3	1.5
FS6	Thailand, anchovy content 55%	0.90 ± 0.09	<LOD	0.8
FS7	Thailand, anchovy content 63%	30 ± 2	33 ± 2	26
FS8	Thailand, anchovy content n.a.%	5.4 ± 0.5	6.4 ± 0.6	4.9
FS9	Thailand, anchovy content 70%	34 ± 3	36 ± 2	29
FS10	Thailand, anchovy content 70%	51 ± 5	52 ± 4	43

*c* = calculated average mass concentration of Hg in fish sauce samples; n.a.: not available.

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
