# Peer review of "Determination of Mercury in Fish Sauces by Thermal Decomposition Gold Amalgamation Atomic Absorption Spectroscopy after Preconcentration by Diffusive Gradients in Thin Films Technique"

_foods, 2020, doi:10.3390/foods9121858_

Round 1

Reviewer 1 Report

General comments

The manuscript by Diviš et al. (foods-1005228) deals with the determination of mercury in fish sauces by Comb-AAS, following preconcentration by DGT technique. The subject of the manuscript is interesting, since Hg monitoring in fish products in significant, while the matrix of fish sauces samples is complex, with considerable salt concentrations, hindering the analysis. Validation of the method was carried out through comparison with direct measurements with Comb-AAS. The authors should improve the 'results and discussion' section, mainly regarding the explanation of several terms mentioned, including figures and tables.

Detailed comments

  • 37-39. The paragraph should be reconsidered, since the consideration 'Fish used in the production of fish sauce are harvested from waters contaminated by varying amounts of metals' is not generally applied.
  • 40-43. The meaning of this paragraph is not clear.
  • 69-76. The aim/scope of the work as well as its novelty should be mentioned more clearly.
  • 101. Add the city of AAS manufacturer.
  • 114. Replace 'AAS' with 'Comb-AAS'.
  • Section 2.3; L.212-222. The CRM ERM-CC580 used is an estuarine sediment and the participation in an interlaboratory comparison includes a wastewater, which are both different from the fish sauces examined. The reasons of their use should be clarified.
  • 263-264. Describe in detail the 'R-value' and ' CSOL'.
  • Table-1. The caption of the table should be modified in order to describe clearly its content. Check the significant digits of the results.
  • Validating a method, the uncertainty and reproducibility could be presented. All data of the method validation (LOD, LOQ, bias, repeatability, reproducibility, uncertainty) should be presented in a separate table.

Author Response

Dear Reviewer No.1, thank you for your valuable comments and suggestions which helped us to improve the quality of the paper. We marked your questions as Q, our answers as A in the following text.

Q: 37-39. The paragraph should be reconsidered, since the consideration 'Fish used in the production of fish sauce are harvested from waters contaminated by varying amounts of metals' is not generally applied.

A: This part of the text has been edited. A new paragraph has been created (lines 37-49).

Q: 40-43. The meaning of this paragraph is not clear.

A: This part of the text has been edited. A new paragraph has been created (lines 37-49).

Q: 69-76. The aim/scope of the work as well as its novelty should be mentioned more clearly.

A: This part of the text has been edited. A new paragraph has been created (lines 70-81)

Q: Add the city of AAS manufacturer.

A: The mention of using the AAS instrument has been removed from this part of the text. A complete description of the device is given in chapter 2.2.

Q: Replace 'AAS' with 'Comb-AAS'.

A: We agree with the modification of the abbreviation, instead of comb-AAS we used TDA-AAS in the text, which is a commonly used abbreviation for thermal decomposition gold amalgamation atomic absorption spectrometry.

Q: Section 2.3; L.212-222. The CRM ERM-CC580 used is an estuarine sediment and the participation in an interlaboratory comparison includes a wastewater, which are both different from the fish sauces examined. The reasons of their use should be clarified.

A: The selection of materials used for AAS validation was additionally described in Chapter 2.1, lines 108-117.

Samples of wastewater and fish muscle were used because their properties are most similar to the analyzed samples in terms of state, or terms of the content of some specific substances (proteins, fats, etc.). We did not find any certified material similar to fish sauce on the market and any similar material available for external quality assessment. The certified sediment sample was used in this study as it represented a completely different type of matrix than previous samples used in method validation. We understand that the use of this CRM may seem to be the most problematic, but this CRM was not used to determine validation parameters presented in Table 1, it was used only for daily testing of the correct function of the instrument before other analyses were started.

Q: 263-264. Describe in detail the 'R-value' and ' CSOL'.

A: The use of the value of R, including the calculation and description of individual variables is already described in the text on lines 179-181. Therefore, this issue is no longer described in lines 263-264. However, we have unified the notation of all variables in the text.

Q: Table-1. The caption of the table should be modified in order to describe clearly its content. Check the significant digits of the results.

A: The caption of the original Table 1 (now 2) has been modified. The results in Table 1 (now 2) were all adjusted to two significant digits.

Q: Validating a method, the uncertainty and reproducibility could be presented. All data of the method validation (LOD, LOQ, bias, repeatability, reproducibility, uncertainty) should be presented in a separate table.

A: All required parameters were summarized in a new Table 1.

Reviewer 2 Report

The authors describe the results of a new type of binding gel containing Purolite S924 resin for application in Diffusive Gradients in Thin Films technique (DGT). After validating the use of DGT with Purolite S924 binding gel, the results are very interesting because provide a gentle and sensitive method for the determination of total dissolved Hg in complex high salt matrices such as fish sauce.
I think that the aim of journal was only partially achieved. Paper focuses too much on validation and little on food safety aspects of fish sauce. The manuscript can be improved with some items indicated below. Some papers could be cited to improve the introduction:
To improve introduction, add and discuss these references:
1. Ariano, A., Marrone, R., Andreini, R., Smaldone, G., Velotto, S., Montagnaro, S., Anastasio, A., (...), Severino, L. Metal concentration in muscle and digestive gland of common octopus (Octopus vulgaris) from two coastal site in Southern Tyrrhenian Sea (Italy) (2019) Molecules, 24 (13), art. no. 2401. https://www.mdpi.com/1420-3049/24/13/2401/pdf;
Explain better the fish sauce sampling and selection procedure.
Check English and References section.

Author Response

Dear Reviewer No.2, thank you for your valuable comments and suggestions related to our article. We marked your questions as Q, our answers as A in the following text.

Q: To improve introduction, add and discuss these references:
Ariano, A., Marrone, R., Andreini, R., Smaldone, G., Velotto, S., Montagnaro, S., Anastasio, A., (...), Severino, L. Metal concentration in muscle and digestive gland of common octopus (Octopus vulgaris) from two coastal site in Southern Tyrrhenian Sea (Italy) (2019) Molecules, 24 (13), art. no. 2401.
https://www.mdpi.com/1420-3049/24/13/2401/pdf;

A: Thank you for linking to an interesting article. Even though the reviewers' request was to shorten the list of used and cited references, we inserted a link to this publication in the newly created chapter 3.4.

Q: Explain better the fish sauce sampling and selection procedure.

A: The primary objective of this study was to verify the functionality of the DGT technique with a new binding gel and to verify whether this technique can be used for the analysis of mercury in foods containing high amounts of salt, such as fish sauce. Therefore, only available samples from shops and Asian markets were purchased. 10 samples were the maximum amount that could be obtained in this way. In the newly created chapter 3.4. it is mentioned that in order to carry out a study with a higher informative value verifying the health risk associated with the consumption of fish sauce, it would be necessary to analyze a wider range of samples.

Q: Paper focuses too much on validation and little on food safety aspects of fish sauce

A: A new chapter 3.4 has been created, (lines 347-361) where the concern for public health is discussed.

Q: Check English and References section.

A: A language correction was made and the citation format was unified.

Reviewer 3 Report

The manuscript concerns an important from the research and application point of view, which is the measurement of mercury concentration in products derived from fish.
The abstract is flawless.
The choice of the analytical method, its performance and validation, shows that the authors are very well prepared for this type of research.
The results are presented logically and correctly. However, in my opinion figures 1 and 2 do not contribute anything and should be omitted, and the information can be included in the text. References are correctly selected However, 50 items seem to be too many for experimental work.

Author Response

Dear Reviewer No.3, thank you for your valuable comments and suggestions related to our article. We marked your questions as Q, our answers as A in the following text.

Q: In my opinion figures 1 and 2 do not contribute anything and should be omitted, and the information can be included in the text.

A: We agree that Figure 2 may be removed and that the results presented in this figure should be described in the text only. The original Figure 2 has been deleted. However, Figure 1 is important because it allows the reader to compare the results we have achieved with the results of other studies dealing with the application of DGT for the determination of mercury. This time series experiment is a basic test performed in every laboratory working with DGT. Since the original Figures 1 and 4 presented the results obtained using the same type of test and only the sample matrix was different, we merged these original figures into one, new Figure 1.

Q: References are correctly selected However, 50 items seem to be too many for experimental work.

A: We have reduced the number of citations to the maximum possible extent, 42 publications are newly cited.

Round 2

Reviewer 2 Report

all requests for improvement have been considered.